# Occurrence and Exposure Assessment of Zearalenone in the Zhejiang Province, China

**DOI:** 10.3390/toxins17010009

**Published:** 2024-12-27

**Authors:** Zijie Lu, Ronghua Zhang, Pinggu Wu, Dong Zhao, Jiang Chen, Xiaodong Pan, Jikai Wang, Hexiang Zhang, Xiaojuan Qi, Qin Weng, Shufeng Ye, Biao Zhou

**Affiliations:** 1School of Public Health, Hangzhou Medical College, Hangzhou 310013, China; 130232023409@hmc.edu.cn (Z.L.); 881012022119@hmc.edu.cn (Q.W.); 2Zhejiang Provincial Center for Disease Control and Prevention, Hangzhou 310051, China; rhzhang@cdc.zj.cn (R.Z.); pgwu@cdc.zj.cn (P.W.); dzhao@cdc.zj.cn (D.Z.); jchen@cdc.zj.cn (J.C.); xdpan@cdc.zj.cn (X.P.); jkwang@cdc.zj.cn (J.W.); hxzhang@cdc.zj.cn (H.Z.); xjqi@cdc.zj.cn (X.Q.); 17867970961@163.com (S.Y.); 3School of Public Health, Ningbo University, Ningbo 315211, China

**Keywords:** zearalenone, precocious puberty, dietary exposure, risk assessment

## Abstract

This study aims to examine the hazards of zearalenone (ZEN) to humans and assess the risk of dietary exposure to ZEN, particularly in relation to precocious puberty in children from the Zhejiang Province. The test results from five types of food from the Zhejiang Province show that corn oil has the highest detection rate of 87.82%. The levels of ZEN do not exceed the existing safety standards in any sample investigated in this study. According to the data from the Food Consumption Survey of Zhejiang Province residents, rice is the primary source of ZEN exposure, accounting for 55.85% of total exposure among all age groups. Based on the 50th exposure percentile, it would take 6.25 years of rice consumption to reach 1 year of safe ZEN exposure. Overall, the majority of the residents in the Zhejiang Province have a low risk of exposure to ZEN. In an extreme case (based on the 95th exposure percentile), the total ZEN exposure from the studied foods with respect to children aged ≤6 years and 7–12 years is 0.38 μg/kg b.w. and 0.26 μg/kg b.w., respectively—both exceeding the safety limit of 0.25 μg/kg b.w. set by the European Food Safety Authority, indicating a potential risk of exposure. Precocious puberty assessments show that ZEN exposure levels in children in the Zhejiang Province are significantly lower than those associated with precocious puberty; thus, precocious puberty is unlikely to occur in this area. Given ZEN’s estrogenic effect, it is necessary to monitor the level of ZEN in different food items, revise the relevant standards as needed, and focus on exposure to ZEN in younger age groups.

## 1. Introduction

Mycotoxins are toxic secondary metabolites produced by fungi and are primarily formed in many cereal crops. Zearalenone (ZEN) is a nonsteroidal mycotoxin produced by *Fusarium* species, specifically *F. graminearum* and *F. semitectum*, and it is similar to 17β-estradiol, the main hormone produced by human ovaries [1,2,3]. It has estrogen-like effects, binds to estrogen receptors, and affects the reproductive system [4]. Currently, five metabolites are known: α-zearalenol (α-ZEL), β-zearalenol (β–ZEL), α-zearalanol (α-ZAL), β-zearalanol (β-ZAL), and zearalenone (ZEN) [5]. ZEN mainly contaminates cereals, most commonly maize, and other crops such as barley, oats, wheat, and rice [6]. ZEN is fat-soluble and almost insoluble in water [7]; this characteristic leads to the accumulation of ZEN, which is not easily eliminated, in the body. ZEN remains stable during storage, has a high melting point, is stable during cooking, and is not easily destroyed by pasteurization [6,8]. Therefore, we should prioritize the control of ZEN production in crops. Given its characteristics, people may be exposed to ZEN through the ingestion of contaminated foods, possibly causing damage to the human body.

ZEN contaminates, to varying extents, cereals and their products worldwide because of their different geographic locations and climatic characteristics. Korean maize and mixed cereal ZEN contamination has been found to reach 4.3 μg/kg and 6.1 μg/kg, respectively, with detection rates of 7% and 47%, respectively [9]. The average contamination of Portuguese and Dutch wheat flour has been estimated at 10.7 μg/kg and 13.1 µg/kg, respectively [10]. Italian wheat and barley contamination have been found to reach 2.4–27.2 μg/kg and 7.3–15 µg/kg (8.8% and 22%), respectively [11]. Other studies have found ZEN detection rates of 11.1% and 66.7% in Brazilian and Mexican maize kernels, respectively [12,13]. Contamination of Algerian maize, rice, and wheat has been estimated at 109 μg/kg, 9.9 μg/kg, and 102 μg/kg, on average, respectively, with the highest detection rate of 63.3% in wheat and no detection in barley [14]. Côte d’Ivoire has been found to have a 100% ZEN detection rate in maize grain and millet grain, with an average contamination of 82.8 μg/kg and 141.9 µg/kg, respectively, and an average detection rate of 80% and 100% in maize and millet-based porridge, respectively [15]. Overall, in previous studies, the African region (Algeria and Côte d’Ivoire) has been found to exhibit higher ZEN detection and contamination rates in cereals and their products than other regions. The unique geographical and climatic conditions of Africa’s tropical region, as well as the lack of proper processing techniques and storage conditions, contribute to the high levels of contamination in African grains [16].

The type of grains, climate, precipitation, storage conditions, and processing affect ZEN amounts in different foods [7,17]. A European Food Safety Authority (EFSA) study has found the highest levels in corn, corn flour, and wheat bran, and relatively high levels in corn germ oil and wheat germ oil [6]. ZEN is a common *Fusarium* mycotoxin in temperate regions, where humid and warm areas favor mycotoxin survival [18,19].

The liver is a major accumulator of ZEN, and exposure to ZEN leads to hepatocellular damage and impaired liver function [7,20]. ZEN impairs the immune function, inhibits T-cell-mediated immune responses by damaging lymphoid organs, and is toxic to the thymus [21]. The genotoxicity of ZEN manifests through the development of chromosomal aberrations, DNA adductions and breaks, and apoptosis [7]. ZEN has been assigned to the carcinogen group 3 based on insufficient evidence from humans and limited evidence from experimental animal studies [22]. Reproductive toxicity is one the main toxic effects of ZEN.

ZEN and its metabolites trigger precocious puberty in children. α-ZAL, a metabolite of ZEN, is used as a beef and lamb synthesis enhancer in the United States and other countries but is banned in the European Union. Additionally, α-ZAL is deemed to be unfit for human consumption within 60 days of its use in animals, including foods of animal origin such as eggs and milk [23,24]. A school in Italy found that both boys and girls had enlarged breasts and had slightly higher estrogen levels in their bodies as a result of ZEN exposure [25]. The same has been reported in Puerto Rico, and most cases occur among girls, who are at a higher risk than boys [24,26]. ZEN can be isolated from the blood of some patients, with the results from two case studies suggesting that the most likely causative factor is the consumption of estrogen-contaminated meat leading to premature breast development [24,25,27].

Owing to the prevalence of ZEN contamination and its multiple toxicities, dietary exposure of the population to ZEN has been evaluated in many countries and regions worldwide. Dietary exposure to ZEN through rice consumption in Pakistan has been estimated at 15.4–23.9 ng/kg body weight (b.w.) and 15.5–24.7 ng/kg b.w. for adults and children, respectively [28]. ZEN exposure through rice consumption of Iranian adults and children has been estimated at 10.86 ng/kg b.w. and 25.34 ng/kg b.w. [29]. The results of the second French total diet study show that dietary exposure to ZEN is in the range of 5.9–25.5 ng/kg b.w. and 11.5–46.2 ng/kg b.w. for adults and children, respectively [30]. Mean and P95 exposures for the Belgian population have been estimated at 0.0447 µg/kg b.w.and 0.1568 µg/kg b.w., respectively [31]. In Panama, the dietary exposure through rice consumption in relation to adults weighing 70 kg has been found to range from 0.03 µg/kg b.w. to 0.07 µg/kg b.w. [32]. However, in relation to Moroccan citizens, ZEN exposure through pasta and green tea consumption has been estimated at 0.0002 µg/kg b.w. and 0.00051 μg/kg b.w. [33,34], respectively. Exposure to maize, rice, and wheat in Algeria has been found to reach 0.08 µg/kg b.w., 0.0013 µg/kg b.w., and 0.85 μg/kg b.w. [14], respectively, and Côte d’Ivoire infants and young children tend to be exposed to 1.14 µg/kg b.w. and 0.46 μg/kg b.w. of ZEN, respectively [15]. Overall, the risk of dietary exposure is low for the residents of most regions, with some residents of African countries being at higher risk of exposure. However, the existing assessment models are homogenous and lack exposure assessments for reproductive toxicity [15,18,35,36,37]. Zhejiang Province is located in the southeastern coastal area of China and has a subtropical monsoon climate. To the best of our knowledge, no studies have assessed the risk of dietary exposure to ZEN among the residents of the Zhejiang Province, China. This study analyzes the status of ZEN contamination across various types of food products in the Zhejiang Province. Exposure assessments are divided into three parts. First, we combine the 50th (P50) and 95th (P95) pollutant concentration and consumption percentiles and use four models for the assessment, an approach which is more accurate than using only one assessment model. Second, we assess the effects of ZEN on humans by calculating the tolerable exposure time using the tolerable daily intake (TDI) specified by EFSA and the population’s probable daily intake (PDI). Third, we assess the risk of precocious puberty caused by ZEN exposure in children.

## 2. Results

### 2.1. Occurrence of ZEN in the Zhejiang Province

Table 1 presents the contamination results from foods marketed in Zhejiang. Non-detected (ND) results are presented as 0 and the limit of detection (LOD) is used to obtain the lower bound (LB) and the upper bound (UB) concentrations, respectively. The national standards for food safety in China, GB 2761-2017, specify a maximum limit of 60 μg/kg of ZEN in wheat, wheat flour, maize, and maize flour only [38]. Due to the limited number of food groups covered by this standard, the relevant EFSA standards were added to the Table 1 [6]. Among the various types of food collected, the contamination levels in all samples were found to be below the established legal maximum limit in China and below the EFSA maximum limit. The chi-squared test show that the detection rates in corn oil (87.82%) and puffed foods (56.45%) were significantly higher than those found in the other three foods, and the detection rate in corn oil was higher than that in puffed foods (*p* < 0.05).

With respect to cereals and their products, the detection rate in cereal supplements for infants and young children was 26.23%, with a maximum level of 15.3 μg/kg, thus not exceeding the maximum limit of 20 μg/kg of ZEN stipulated by the Zhejiang Agricultural Products Quality and Safety Society [39]. All cereals and their products exhibited ZEN levels below the legal limit.

With respect to noodle products, the ZEN detection rate was 13.16% in instant noodles, which exhibited the highest ZEN concentration of 12.4 μg/kg, with ZEN not detected in either sample of wet noodle products. Currently, neither China nor the EFSA have any regulations regarding the limits of ZEN in noodle products. Noodles are derived from wheat flour, and none of the samples evaluated in this study exceeded the maximum limit of 60 μg/kg in wheat flour set by Chinese standards.

With respect to the three types of tea for which samples were collected, positive rates were higher in the green tea, at 20%, with positive rates attaining 9.1% in both dark and black tea. Positive rates were 56.45% and 87.82% in puffed food and corn oil, respectively, and the highest percentage of positive samples was found in the corn oil; the highest level detected was 371 μg/kg. Currently, there is no restriction on the content of ZEN in corn oil, tea, and puffed food in China, with the EFSA limit for ZEN in refined corn oil being 400 μg/kg [6]. None of the samples taken exceeded this standard.

### 2.2. Consumption of Surveyed Foods

Table 2 presents the food consumption rates of the different population groups. Different populations may have different dietary patterns, and, for a more accurate assessment, we considered five different age groups: children (≤6 years), older children (7–12 years), adolescents (13–17 years), adults (18–59 years), and the elderly (≥60 years). Because of the lack of consumption data in relation to corn oil, dried noodle products, puffed food products, and cereal supplements for infants and young children, the data from other relevant investigations and from the National Bureau of Statistics served as the mean consumption data of cereal supplements for infants and young children. The following consumption rates were found with respect to corn oil (30% of vegetable oil consumption), puffed food, and dry noodle products: 28.3 g/d, 8.22 g/d, 9.76 g/d, and 3.84 g/d, respectively. An average body weight of 60 kg was assumed for adults, while a weight of 10.1 kg was assumed for children aged ≤36 months [40,41,42,43].

### 2.3. Dietary Exposure Assessment

#### 2.3.1. Exposure Assessment Based on Different Age Groups Under Different Scenarios

The dietary exposure assessment of ZEN mostly relies on the point-assessment method. Exposure to mycotoxins through food consumption was first assessed in subgroups based on age using a deterministic approach (Table 3). Despite the results of existing studies suggesting a low risk of dietary exposure, the foods examined in this study represent only a portion of the population’s diet.

Scenario 1 was calculated using P95 consumption data and P95 contamination levels, both of which represent extreme-case exposure levels. Based on age, the children (≤6 years) were the group with the highest exposure values. In Scenario 1, the dietary exposure value for rice was 0.360 µg/kg b.w. in children, indicating that children (≤6 years) who consume large amounts of rice are at risk of ZEN exposure. Among the analyzed food groups, the highest exposure risk was associated with rice intake (55.85%).

The index of food safety (IFS) is another dietary exposure assessment method in which the risk assessment is performed by calculating the ratio of the estimated intake to the safe intake. In Scenario 1, the IFS of rice consumption for children aged 6 years and younger was 1.44, which exceeds the acceptable limits and indicates a high risk of exposure. In the other scenarios, the IFS of all types of food consumed by people of all ages was less than 1, which is within the acceptable range.

In Scenario 1, the total PDI for ZEN showed the following trend: children (≤6 years) > older children (7–12 years) > adults (18–59 years) > teenagers (13–17 years) > elderly (≥60 years), with the total ZEN exposure in children aged ≤6 years and 7–12 years being 0.38 μg/kg b.w. and 0.26 μg/kg b.w., respectively. This is higher than the TDI of 0.25 μg/kg b.w. established by the EFSA [6]. Thus, children (≤6 years) and older children (7–12 years) experience greater exposure to ZEN than adults and the elderly, and, in extreme cases, children are at risk of exposure. Therefore, increased attention should be paid to the health risks posed by ZEN in foods which are consumed by children; however, little information has been reported on the risk assessment of ZEN in cereal supplements for infants and young children.

#### 2.3.2. ZEN Non-Carcinogenic Exposure Assessment

Table 4 presents the duration of tolerance to non-carcinogenic ZEN in the Zhejiang population. The P50 consumption data and P50 contamination level data were used, and the tolerable durations of exposure to ZEN were estimated at 6.25 years, 215,045.83 years, 75,071.20 years, 6467.53 years, 202,193.74 years, and 36,879.62 years for rice, millet, maize and its products, wet noodle products, instant noodles, and tea, respectively. Tolerable exposure duration was as follows: rice < wet noodle products < tea < maize and its products < instant noodles < millet. The food with the shortest tolerable exposure time was rice, indicating that it would take 6.25 years of eating rice to consume a year’s worth of ZEN. The remainder of the foods took thousands or even tens of thousands of years to reach the safe exposure of one year. Therefore, the exposure risk based on ZEN intake among the residents of the Zhejiang Province is low.

## 3. Discussion

We investigated the current ZEN contamination status in foods from the Zhejiang Province and assessed dietary exposure among different age groups by combining data on food consumption with data on contaminant concentrations in the food. None of the food types studied exceeded the limits. Most of the population was associated with a low risk of exposure, and children were associated with some risk of exposure based on the P95 exposure assessment; however, such a risk of exposure was significantly lower than the exposure dose required for precocious puberty.

Corn oil had the highest ZEN detection rate in this study, such a finding being similar to the results from an EFSA study [6]. In this study, the detection rate of ZEN in marketed corn oil from the Zhejiang Province was 87.82%, a value higher than that documented in the Shandong Province (detection rate: 72.1%, maximum concentration: 437.4 µg/kg) [44] and in Tianjin (70%, 90.92 µg/kg) [45]. A German study on corn germ oil has shown an average contamination level of 169 µg/kg and a maximum contamination of 921 µg/kg [46]. Maize is highly susceptible to contamination, and, when maize and maize germ are stored improperly, they can easily become contaminated with ZEN, which then enters maize oil through the pressing process [6,47]. To produce corn oil, the contaminated corn is pressed until it becomes liquid and highly fluid, resulting in corn oil being contaminated with ZEN and leading to a high detection rate. The level of ZEN contamination in corn oil may vary depending on the region, raw material, and processing method [48]. Although the detection rate in corn oil is high, its associated risk of ZEN exposure is low owing to its low consumption rates, even assuming a cooking oil consumption rate of 27.4 g/day per person.

An EFSA’s study has shown that ZEN is primarily found in various cereals and their products [6]. In this study, the detection rate in cereal supplements for infants and young children was 26.23%, higher than that documented in other provinces of China [49,50,51,52]. In another study on the consumption of rice flour by infants in China, the ZEN detection rate has been estimated at 1.4%, with ZEN detected in all imported products [53]. The detection rate in baby food in Spain has been found to reach 23.3%, with an average ZEN concentration of 4.1 µg/kg [54]. The detection rate in Canada has been estimated at 32.6%, with the highest contamination level being 35 µg/kg [55]; no ZEN contamination was detected in any of the samples collected from the German market [56]. A few studies have examined the levels of ZEN in cereal-based complementary foods for infants and young children, and dietary exposure assessments I this context are lacking. As ZEN remains stable during processing [6], high contamination of raw materials can affect the contamination levels in the food. Damage to cereal hulls during storage and processing should be avoided, and the temperature and humidity of the storage environment should be controlled to minimize toxin production [57].

The remaining cereals and their products were contaminated to varying degrees in this study. The ZEN detection rate in maize and its products in the Zhejiang Province was 19.35%. The highest concentration of ZEN to ever be found in maize in Portugal is 900 μg/kg, higher than the average concentration of 70 μg/kg found in all cereal samples tested in this study [58]. The detection rate in Korean maize flour has been estimated at 88%, which is higher than that in wheat flour (33%), rice flour (20%), and oat flour (5%) [59]. These results show that maize is more contaminated with ZEN than other cereals. In a study conducted in Henan, China, the ZEN content of corn kernels (18.8%, 39.24 μg/kg), corn residue (29.7%, 336.48 μg/kg), and cornmeal (40.4%, 347.4 μg/kg) was examined separately [60], suggesting that ZEN in fresh corn may be lower than in corn products. Similar findings have been reported for other regions [61,62]. The average level of ZEN in corn samples from California has been estimated at 10.31 μg/kg, with even lower levels reported in fresh corn [63]. The processing method affects the ZEN content of cereal products. This is the reason behind the higher detection of ZEN in cereal-based puffed food products (56.45%) in the Zhejiang Province.

Based on the results from the dietary exposure assessment, the exposure contribution of rice is higher, probably because rice is a staple food among the residents of the Zhejiang Province. In general, the risk of exposure to ZEN among the residents of most areas in the Zhejiang Province is low. In Scenario 1, the exposure to ZEN through rice consumption was relatively high in children aged ≤6 years in China, and the exposure to ZEN from rice consumption in children aged 7–12 years was slightly below the critical value, a finding which must be emphasized. Reproductive toxicity is the main hazard of ZEN and is associated with precocious puberty.

The results from the dietary exposure assessment combining pollutant concentration and consumption data show that the risk of ZEN exposure is low among the residents of the Zhejiang Province. The dietary exposure risk in children is higher than in other groups. The PDI of maize and its products consumed by Chinese people has been estimated at 5.1–5.47 ng/kg b.w. [64], and the average ZEN intake of adults in Hong Kong has been shown to reach 0.0061–0.1015 μg/kg b.w. [35]. Other countries have also assessed dietary exposure to ZEN. Dietary ZEN exposure through the consumption of various cereals and their products has been estimated at 0.005 μg/kg b.w., 0.012 μg/kg b.w., and 0.075 μg/kg b.w. in Brazilians, Iranians, and Italians, respectively [11,12,36]. Spanish infants and toddlers have been found to experience ZEN exposures of 12.2–17.9 ng/kg b.w. [54]. The population in most countries and regions is subjected to varying degrees of exposure, with the overall risk of exposure being low; however, a small number of countries may be at risk of exposure in extreme cases.

We combined rat experiments with population studies to assess precocious puberty in children. Rats are important model organisms in medical research. Yang et al. studied the effects of ZEN on precocious puberty in female rats and found that the no observed adverse effect level (NOAEL) of ZEN on estrogenic activity in immature female rats was 0.2 mg/kg b.w., while the lowest observed adverse effect level (LOAEL) was 1 mg/kg b.w. [65]. An uncertainty factor of 100 was used to estimate the exposure to estrogenic activity and puberty promotion in infants and young children under a concentration of 2 μg/kg b.w. [66]. Owing to the limited references from animal experiments, we compared our results with those obtained from urine exposure assessments in children with precocious puberty. Based on a study by Asci et al. on ZEN levels in the urine of the control, premature thelarche (PT), and central idiopathic precocious puberty (ICPP) groups, the PDIs for the three groups, estimated using the formula for urine exposure assessment, were 2.2 μg/kg b.w., 4.1 μg/kg b.w., and 6.1 μg/kg b.w., respectively [67]. No risk of precocious puberty was observed at exposure levels of 2.2 µg/kg b.w., a risk of PT was found at exposure levels above 4.1 µg/kg b.w., and a risk of ICPP was found at exposure levels above 6.1 µg/kg b.w., while the PDI of the control group was close to the 2 µg/kg b.w. concentration obtained in the rat experiments. In conclusion, precocious puberty in children from the Zhejiang Province exposed to ZEN through ingestion of the study foods is considered to be unlikely, even in extreme cases. Controversy exists regarding the carcinogenicity of ZEN. Based on the results from available studies, the concentrations of ZEN in the blood or urine can be used to measure exposure to assess the risk of developing cancer [68,69,70,71]. Khosrokhavar et al. found that a certain concentration of ZEN had a stimulatory effect on the growth of breast cancer cells, an inhibitory effect at elevated concentrations, and the metabolite of ZEN, α-ZAL, was more stimulatory to breast cancer cells [68]. The concentration of ZEN and its metabolites is higher in endometrial cancer cells than in normal endometrial cells [71,72,73]. Combined with the results from available studies and references, its endocrine-disrupting effects may also be a predisposing factor for carcinogenesis, suggesting that ZEN plays a potential role in estrogen-related carcinogenesis. This study focused on the non-carcinogenic effects of ZEN. Based on the tolerable exposure time results, rice has the shortest one-year tolerable exposure time of 6.25 years, and the remaining food items have a greater tolerable exposure time than rice because of their low consumption rates.

This study has some limitations. First, the food types collected in this study were limited. Second, the consumption data used in this study were from 2015 to 2017, and the detection of pollutant concentrations was conducted from 2018 to 2019. Finally, the risk of ZEN may be underestimated because we did not consider the metabolites of ZEN.

## 4. Conclusions

Food products from the Zhejiang Province show different levels of ZEN contamination, but none exceeds the current safety standards. Although the ZEN detection rate in corn oil is high, its associated risk of exposure is low because of its low consumption rates. According to the point assessment results, the risk of dietary exposure to ZEN is low among most residents in this study; however, children aged ≤6 years are at some risk of exposure from rice consumption under scenario 1, possibly affecting their growth and development adversely. Based on the duration of tolerance to non-carcinogenic ZEN, the shortest tolerable exposure time for rice intake is 6.25 years; therefore, more attention needs to be paid to the ZEN content in rice, although the exposure risk is low. The results of the risk assessment for precocious puberty show that the risk of developing precocious puberty due to exposure to ZEN in children living in the Zhejiang Province is low.

## 5. Material and Methods

### 5.1. Sample Collection

The collected samples included cereals and their products (*n* = 278, containing rice, maize and its products, millet, oats, brown rice, cereal supplements for infants and young children), noodle products (*n* = 97, containing instant noodles, dry noodle products, and wet noodle products), tea (*n* = 74, containing dark tea, black tea, and green tea), puffed foods (*n* = 62, containing shrimp chips, snow cakes, and rice crisper candy), and corn oil (*n* = 197). The samples required for this study were randomly collected by professionals from various regions of the Zhejiang Province from 2018 to 2019, and 708 samples were collected. The sample size was calculated using the following formula [74,75]:(1)$$N=\frac{Z^{2}\times \left[P\times \left(1-P\right)\right]}{e^{2}}$$
where *N* is the sample size, *Z* = 1.96 is the level of confidence (95%), *P* = 0.5 is the expected contamination rate, and *e* = 10% is the margin of error. We collected 708 samples in the Zhejiang Province. The sampling sites primarily included supermarkets of different sizes, farmers’ markets, and online shopping platforms. After collection, we sent the sample to the monitoring technical institution or laboratory for testing under conditions close to the original storage temperature.

### 5.2. Preparation of the Standard Solution

Pure mycotoxins of ZEN and the ^13^C_18_-ZEN isotope internal standard solution (25 µg/mL) were purchased from Sigma-Aldrich (St. Louis, MO, USA). A given amount of ZEN was dissolved in acetonitrile to obtain a single standard solution with a concentration of 10 µg/mL. We took 1 mL of ^13^C_18_-ZEN isotope internal standard solution with a concentration of 25 µg/mL and mixed it with a single standard solution to obtain a mixed isotope internal standard working solution with a concentration of 1.0 µg/mL. We mixed the solution well and stored it at −20 °C until chemical analysis.

### 5.3. Sample Preparation and Analysis

The methods and operational procedures used to detect ZEN in food were obtained from the National Risk Monitoring Workbook for Food Contamination and Harmful Factors, published by the China National Center for Food Safety Risk Assessment. The samples were pulverized (0.5–1 mm) using a high-speed pulverizer (100 g per sample) and stored in a sealed container. We took 2 g of the sample and mixed it with 200 µL of the mixed isotope internal standard solution. We then added 10 mL of acetonitrile/water with acetic acid (84/1/16, *v*/*v*/*v*), mixed the solution thoroughly, and centrifuged it at 8000 rpm for 5 min. The filtrate was extracted, put into a nitrogen flask, and blown dry at 50 °C, and the obtained material was redissolved in 1 mL of a 20% acetonitrile–water solution. The solution was passed through a 0.22 µm polytetrafluoroethylene filter membrane for subsequent analysis.

Samples were analyzed using an 8060-LC-MS/MS instrument equipped with an electrospray ionization (ESI) source (Shimadzu, Japan). Chromatographic separation was performed with a BEH C_18_ column (2.1 × 100 mm I.D., 1.7 µm, Waters, Milford, MA, USA) at a column temperature of 40 °C and a flow rate of 0.3 mL/min. The mobile phases A and B were 0.01% ammonia and acetonitrile, respectively.

### 5.4. Method Validation and Quality Control

Researchers involved in the testing passed the training and examinations required to conduct testing at the NHC Specialty Laboratory of Food Safety Risk Assessment and Standard Development. The testing steps and procedures were based on the National Risk Monitoring Workbook for Food Contamination and Harmful Factors, with an entire process and multi-departmental quality control strategy. Methodological validation, i.e., precision, linearity, and recovery, was performed before use. The recoveries and precisions were 100 × (calculated using the calibration curve/spiked concentration). Owing to the large number of food groups in this study, the LOD ranged from 1 µg kg^−1^ to 5 µg kg^−1^ depending on the food group (LOD of 5 µg/kg^−1^ for rice and cereal supplements for infants and young children, 2 µg/kg^−1^ for all other cereals; LOD of 5 µg/kg^−1^ for wet noodle products and 1 µg/kg^−1^ for dry noodles and instant noodles; LODs of 1 µg kg^−1^ and 5 µg/kg^−1^ for tea and corn oil, respectively). The value of the non-detected samples was assumed to be the LOD for dietary exposure.

### 5.5. Food Consumption Data

This study used the food frequency method to conduct a survey in 10 counties (cities) in the Zhejiang Province from 2015 to 2017. The resident population (i.e., individuals with at least 6 months of residence) of the area was the survey population. The survey respondents signed an informed consent form, and all personal information was confidential. For the missing data, refer to the relevant data from the literature (see Section 2.2) for details.

Training enumerators collected information on food consumption, production, processing methods, and consumption habits through household surveys in which the respondents were asked about the type, frequency of consumption, and quantity consumed in relation to a particular food item over a specified period. They also provided information about the individual, such as age, ethnicity, marital status, education, and occupation.

### 5.6. Dietary Exposure Assessment

#### 5.6.1. Dietary Exposure

A deterministic point assessment methodology was used to calculate the PDI by combining pollutant concentrations and food consumption data to estimate the exposure to ZEN in the population of the Zhejiang Province [76]. PDI was calculated using the following formula [30]:(2)$$PDI=\frac{Contamination\;concentrations\;of\;ZEN\;in\;food\;({\mu g/kg}^{-1})\times Consumption\;of\;food\;(g/d^{-1})}{Body\;weight\;(kg)}\times 10^{-3}$$

The point assessment utilized six assessment models, with Scenarios 1, 2, 3, and 4 representing the P95 pollutant concentration and P95 consumption, P50 pollutant concentration and P95 consumption, P95 pollutant concentration and P50 consumption, and P50 pollutant concentration and P50 consumption, respectively. Scenarios 5 and 6 combined the average consumption and average pollutant concentrations and calculated the mean exposure at the population level. Multiple assessment models can be used to account for extreme cases and populations.

The IFS approach quantifies the extent of consumer health risks associated with chemical contaminants in food, calculated as the ratio of the actual intake to the safe intake, to evaluate the impact of ZEN residues in food on the human body. IFS ≤ 1 indicates that chemical contaminants are acceptable, IFS > 1 indicates that the food safety risk of chemical contaminants exceeds the acceptable limit [77,78].

#### 5.6.2. Urine Exposure Assessment

Presence of biomarkers in the urine allows for the estimation of potential human intake, and PDIs estimated using the urine biomarker method align with those obtained by combining consumption and pollutant concentration data. The PDI in the population was estimated by measuring the excretion concentration of ZEN in the urine, combined with the urine volume and the excretion rate. The following formula was used for the calculation [79]:(3)$$PDI=C\times \frac{V}{W}\times \frac{100}{E}$$

In this equation, *PDI* represents the probable daily intake (µg/kg b.w.), *C* represents the concentration of ZEN in the urine (µg/L), *V* denotes the 24 h urine volume of the child (0.8 L) [80], the creatinine value was taken as 1 g/L [81]. *W* indicates the mean weight of the children aged 7–12 years (33 kg), and *E* indicates the excretion rate of ZEN through urination (9.4%) [82].

#### 5.6.3. Tolerable Duration of Exposure to Non-Carcinogenic ZEN

The target hazard quotient (THQ) is a composite risk index proposed by the U.S. Environmental Protection Agency (EPA) that compares the intake of a pollutant with a standardized reference dose [83]. THQ is calculated as follows [84,85]:(4)$$THQ=\frac{C\times FIR\times EF\times ED}{BW\times AT\times Rfd}\times 10^{-3}$$

The THQ indicates the human health risk level due to exposure to pollutants, where *C* denotes the concentration of ZEN in the food (μg/kg), *FIR* represents the daily consumption rate (g/person/day), *EF* denotes the exposure frequency, *ED* indicates the exposure duration, *BW* represents the average body weight (60 kg), *AT* is the average time for noncarcinogens (365 days/year × ED), and *RFD* is the oral reference dose (μg/kg/day). We used a TDI of 0.25 μg/kg b.w., as specified by EFSA, instead of the RFD. Higher THQ values indicate a higher likelihood of long-term non-carcinogenic effects. If THQ < 1, the exposed population is unlikely to experience obvious adverse effects. If THQ ≥ 1, there is a potential health hazard [86]. We assumed THQ = 1 and used a TDI of 0.25 μg/kg b.w. divided by the frequency of exposure to each food to derive a sustainable amount of exposure per year.

### 5.7. Statistical Analysis

The IBM SPSS software (version 25.0; IBM Corp., Armonk, NY, USA) was used for all statistical analyses. Measurements without a normal distribution are expressed as the 50th and 95th percentiles. A chi-squared test was used to determine whether there were differences in the detection rates of ZEN across different types of food. Statistical significance was set to *p* < 0.05.

## Figures and Tables

**Table 1 toxins-17-00009-t001:** Prevalence of ZEN in various food products from the Zhejiang province (*n* = 708).

Food Group	Food Name	Sample Size	Detection Rate	Mean		Max (μg/kg)	Value Limitation (μg/kg)
				LB ^1^	UB ^2^		EFSA ^3^
Cereals and their products	Rice	93	18.28%	3.39	7.48	39.00	-
	Maize and its products	31	19.35%	1.10	2.71	13.20	100
	Millet	31	29.03%	1.44	2.86	7.23	75
	Oats (flakes)	31	3.23%	0.22	2.16	6.83	75
	Brown rice	31	22.58%	1.35	2.90	17.20	75
	Cereal supplements for infants and young children	61	26.23%	2.12	5.81	15.30	20
Noodle products	Instant noodles	38	13.16%	0.68	1.55	12.40	-
	Dry noodle products	57	10.53%	0.66	1.56	11.70	-
	Wet noodle products	2	0%	0.00	5.00	5.00	-
Tea	Dark tea	33	9.10%	1.63	2.54	19.40	-
	Black tea	11	9.10%	0.42	1.33	4.60	-
	Green tea	30	20.00%	2.82	3.62	18.60	-
Puffed food	Puffed food product	62	56.45% *	6.37	8.55	63.20	-
Oil	Corn oil	197	87.82% *	66.97	67.58	371.00	400

^1^ LB, ND = 0; ^2^ UB, ND = LOD; ^3^ EFSA, European Food Safety Authority limits for ZEN; * Indicates statistical significance.

**Table 2 toxins-17-00009-t002:** Daily consumption of grains and grain products among the participants.

Food Name	Consumption (g/day)											
	≤6		7–12		13–17		18–59		≥60		All	
	P50	P95	P50	P95	P50	P95	P50	P95	P50	P95	P50	P95
Rice	120.00	300.00	160.00	360.00	200.00	450.00	240.00	600.00	240.00	480.00	225.00	600.00
Millet	1.00	10.64	0.67	14.29	0.83	11.43	1.00	17.14	1.64	23.21	1.00	14.29
Maize and its products	2.00	16.51	2.67	21.43	2.67	28.57	3.00	25.71	3.00	28.57	3.86	25.71
Wet noodle products	4.28	35.71	5.33	42.86	6.67	50.00	6.67	51.43	6.67	42.86	6.67	48.57
Instant noodles	2.67	14.00	2.77	22.86	3.33	31.07	1.80	17.14	1.48	28.57	2.17	22.86
Tea	0.07	1.04	0.17	0.87	0.14	12.50	0.86	10.00	2.00	13.49	1.00	10.00

**Table 3 toxins-17-00009-t003:** Exposure contribution and dietary exposure to ZEN in different age groups under different scenarios.

Food Name	Scenario	Dietary Exposure (µg/kg b.w.)					Exposure Contributions (%)
		≤6	7–12	13–17	18–59	≥60	All
Rice	Scenario 1 ^1^	*0.360* ^7^	0.249	0.200	0.226	0.188	55.85
	Scenario 2 ^2^	0.078	0.054	0.043	0.049	0.041	
	Scenario 3 ^3^	0.144	0.111	0.089	0.091	0.094	
	Scenario 4 ^4^	0.031	0.024	0.019	0.020	0.020	
Millet	Scenario 1	0.004	0.003	0.001	0.002	0.002	0.34
	Scenario 2	0.001	0.001	<0.001	0.001	0.001	
	Scenario 3	<0.001	<0.001	<0.001	<0.001	<0.001	
	Scenario 4	<0.001	<0.001	<0.001	<0.001	<0.001	
Maize and its products	Scenario 1	0.005	0.003	0.003	0.002	0.003	0.54
	Scenario 2	0.002	0.001	0.001	0.001	0.001	
	Scenario 3	0.001	<0.001	<0.001	<0.001	<0.001	
	Scenario 4	<0.001	<0.001	<0.001	<0.001	<0.001	
Wet noodle products	Scenario 1	0.009	0.006	0.005	0.004	0.004	1.81
	Scenario 2	0.009	0.006	0.005	0.004	0.004	
	Scenario 3	0.001	0.001	0.001	0.001	0.001	
	Scenario 4	0.001	0.001	0.001	0.001	0.001	
Instant noodles	Scenario 1	0.003	0.003	0.002	0.001	0.002	0.22
	Scenario 2	0.001	0.001	0.001	<0.001	<0.001	
	Scenario 3	0.001	<0.001	<0.001	<0.001	<0.001	
	Scenario 4	<0.001	<0.001	<0.001	<0.001	<0.001	
Tea	Scenario 1	0.001	<0.001	0.004	0.003	0.004	0.19
	Scenario 2	<0.001	<0.001	<0.001	<0.001	<0.001	
	Scenario 3	<0.001	<0.001	<0.001	<0.001	0.001	
	Scenario 4	<0.001	<0.001	<0.001	<0.001	<0.001	
Puffed food products	Scenario 5 ^5^	0.001					2.11
Dry noodle products	Scenario 5	<0.001					0.15
Corn oil	Scenario 5	0.009					14.06
Cereal supplements for infants and young children	Scenario 6 ^6^	0.016					24.72

^1^ Scenario 1: P95 contamination levels in the samples combined with P95 daily consumption and average weight of the investigators. ^2^ Scenario 2: P50 contamination levels in the samples combined with P95 daily consumption and average weight of the investigators. ^3^ Scenario 3: P95 contamination levels in the samples combined with P50 daily consumption and average weight of the investigators. ^4^ Scenario 4: P50 contamination levels in the samples combined with P50 daily consumption and average weight of the investigators. ^5^ Scenario 5: mean contamination level in the samples combined with the mean daily consumption and estimated population weight of 60 kg. ^6^ Scenario 6: mean contamination level in the samples combined with the mean daily consumption and average weight of the investigators, set to 10.1 kg. ^7^ Values in italics indicate that the safety standards have been exceeded.

**Table 4 toxins-17-00009-t004:** Tolerable durations of exposure to non-carcinogenic ZEN according to the consumption and contamination data.

Food Name	P50 Exposure ^1^ (UB ^2^, µg/kg b.w.)	Exposure Duration (Years)
Rice	0.0200	6.25
Millet	<0.0001	215,045.83
Maize and its products	0.0001	75,071.20
Wet noodle products	0.0006	6467.53
Instant noodles	<0.0001	202,193.74
Tea	<0.0001	36,879.62

^1^ P50 consumption with P50 contamination level. ^2^ UB, ND = LOD.

## Data Availability

The data presented in this study are available on request from the corresponding author. The data are not publicly available due to privacy.

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
