# Peer review of "Occurrence and Exposure Assessment of Zearalenone in the Zhejiang Province, China"

_toxins, 2024, doi:10.3390/toxins17010009_

Round 1

Reviewer 1 Report

Comments and Suggestions for Authors

I read the manuscript entitled “Occurrence and Exposure Assessment of Zearalenone in Zhejiang Province, China”. The paper is captivating and scientifically sound. This article is super interesting and informative for the Chinese community. However, very minor errors were detected.

Abstract

Overall, the abstract wording is inappropriate and misleading, weakening the science.

Line 6: The phrase "residing in Zhejiang Province" could be more concise. Consider rephrasing it to "in children from Zhejiang Province."

Line 10: The phrase "the shortest sustainable exposure time" is slightly unclear. Consider rephrasing it to something like "the minimum continuous exposure duration."

Line 11-12- Based on P50 exposure…………………………..rice was 6.25 years. It is a confusing statement, it might be %.

Line 14: The term "extreme case" is vague. Clarifying what defines an extreme case (e.g., highest consumption levels or specific populations) would help the reader understand the context better.

Lines 13–14: The use of "total exposure to ZEN for children ≤6 years and 7–12 years" could be clarified. It’s unclear whether this refers to an average or maximum value for the group.

Line 9: The statement that rice was the "primary source of ZEN exposure" contributing 55.85% is crucial but could use additional context. Was this the case across all age groups, or specific to certain populations?

Line 16: The mention of the "combined rats' experiment" is helpful but could be confusing to some readers. It would be clearer to briefly explain why the rat model is relevant to human exposure and risk.

Introduction  

The paragraph beginning with "Owing to the prevalence of ZEN..." (line 96) introduces dietary exposure assessments without sufficient connection to the earlier points. Consider a clearer transition to emphasize why these studies are relevant to your research.

Line 33- Ensure consistency when mentioning ZEN’s metabolites. Sometimes they are written as “α-ZAL” and “β-ZAL”, and other times as "α-zearalanol" and "β-zearalanol" (line 33). Choose one format and use it consistently throughout the manuscript.

Line 35- The section on ZEN’s solubility could be expanded slightly to explain the practical implications of these properties, especially in the context of food contamination.

Lines 37-38- The stability of ZEN could be further clarified. What are the practical consequences of its resistance to cooking and pasteurization for consumers and food safety regulations?

Lines 68-71-The section on the impact of grain processing could be expanded slightly. For example, what specific processing techniques contribute to reducing ZEN levels, and how prevalent are these techniques in Zhejiang Province?

Discussion

Line 245- The phrase "children had some risk of exposure" could benefit from a more specific quantification or clearer explanation of what constitutes "some risk" for the readers.

Line 261- When introducing EFSA's findings, use consistent phrasing throughout the paper to avoid repetition, such as "EFSA's study showed" in multiple places. This would make the structure more engaging.

Line 343- Cite more recent studies or include a note on whether the contamination levels have changed post-2019, if data are available.

Reviewer 2 Report

Comments and Suggestions for Authors

In manuscript "Occurrence and Exposure Assessment of Zearalenone in Zhejiang Province, Chine" very interesting study was covered but it should be better presented.

In the whole manuscript spaces and italic for Latin names of species should be checked. English language should be improved.

Introduction

Lines 44 and 46 Units should be uniform (ng/g and mg/kg)

Lines 42-57 Many different studies and concentrations of ZEN were mentioned and it is not easy to follow all numbers. Maybe it would be easier to understand this section if those data are presented in table.

Line 72 First sentence in unnecessary because effect on humans are explained in details in the following section.

Lines 84-95 Sentences in the whole section are not well connected. Italy and Puerto Rico cases are just mentioned without explanation. It is not clear whether the following sentence regarding ZEN is just another fact or is it explanation for premature puberty in Italy and Puerto Rico.

Line 113 "…Few studies have assessed…" Please, provide, references.

Line 116 P50 and P95 should be briefly explained. If someone is not familiar with this statistical method, it is impossible to understand the point.

Results

Line 164 Average body weight for infants and young children is 10.1 kg. Is this value calculated from your data or just an assumption? Children younger than 6 years can weight from 3 to 20 kg.

Line 216 Figure 1 Comparison - figure and abbreviation should be explained better. It is not clear what was compared to what, and what are the literature data or data from your research.

Line 239 Table 4 Please, provide better explanation of the table.

Discussion

My general impression of discussion is that there are to many literature facts but explanation is missing. Please, try to connect presented data and explain the importance of presented data from the literature for presented study.

Material and Methods

Line 364 Sample Collection I suggest you start this section with type of samples. It will be easier to follow.

Line 394 Have you prepared samples before adding internal standards?

Line 396 Please, provide parameters for centrifuge.

Line 398 Please, provide the type of filter membrane.

Section 5.4. Please, write detail information for LOD and LOQ for different type of food.

Section 5.6.2. Have you measured metabolites of ZEN in urine?

Line 455 have you measured creatinine in urine?

Line 456 "mean weight of children" - Is it assumption or mean value from your data?

Comments on the Quality of English Language

English language should be improved.

Reviewer 3 Report

Comments and Suggestions for Authors

The article submitted to Toxins for publication on Occurrence and Exposure Assessment of Zearalenone in Zhejiang Province provides data on the presence of ZEN in food and population intake. However, some aspects need to be modified or clarified.

- In the introduction, the existing legislation in China on the presence of ZEN in food should be specified in the text or in table 1. If there is no legislation, comment that this is a serious deficiency and add a column with the EU values ​​in table 1.

- In the introduction, delete the paragraphs between lines 34-41, 62-71 and 77-83 as they are superfluous or unrelated to this article.

- The foods were sampled in 2018 and 2019. When were they analysed? And how was their conservation until analysis?

- No experiments have been performed on rats and no urine analysis of consumers has been carried out, therefore all reference to these should be deleted, for example, delete section 2.3.2, line 15, line 219 and paragraph between lines 308-327.

- No analyses of ZEN metabolites in food were carried out, therefore the presence of ZEN is underestimated and so is the risk, which should be included as a deficiency of the study in paragraph 341-346. Consequently, it should be qualified and named only as "unlikely" instead of "low risk" with the appearance of precocious puberty.

Round 2

Reviewer 3 Report

Comments and Suggestions for Authors

Delete at the bottom of table 1 the paragraph “National 120 standards for food safety in China, GB 2761-2017, specify a maximum limit of 60 μg/kg for ZEN only in wheat, wheat flour, maize, and maize flour) and EFSA standards are added in the table.” As it is sufficiently explained in section 2.1.

- Delete the paragraph “The inspectors were professional technicians trained as required and performed inspections according to the requirements of the monitoring program and workbook.” (lines 336-337) because it is obvious and line 325 already refers to the qualification of the staff.
